# Efficacy of Low-Dose Prophylactic Quetiapine on Delirium Prevention in Critically Ill Patients: A Prospective, Randomized, Double-Blind, Placebo-Controlled Study

**DOI:** 10.3390/jcm9010069

**Published:** 2019-12-27

**Authors:** Youlim Kim, Hyung-Sook Kim, Jong Sun Park, Young-Jae Cho, Ho Il Yoon, Sang-Min Lee, Jae Ho Lee, Choon-Taek Lee, Yeon Joo Lee

**Affiliations:** 1Division of Pulmonary, Allergy and Critical Care Medicine, Department of Internal Medicine, Hallym University Chuncheon Sacred Heart Hospital, Chuncheon-si 24253, Gangwon-Do, Korea; weilin810707@gmail.com; 2Department of Pharmacy, Seoul National University Bundang Hospital, Seongnam-si 13620, Gyeonggi-Do, Korea; kehese2956@snubh.org; 3Division of Pulmonary and Critical Care Medicine, Department of Internal Medicine, Seoul National University Bundang Hospital, Seongnam-Si 13620, Gyeonggi-Do, Korea; jspark.im@gmail.com (J.S.P.); lungdrcho@gmail.com (Y.-J.C.); dextro70@gmail.com (H.I.Y.); jhlee7@snubh.org (J.H.L.); ctlee825@gmail.com (C.-T.L.); 4Division of Pulmonary and Critical Care Medicine, Department of Internal Medicine, Seoul National University Hospital, Seoul 03080, Korea; sangmin2@snu.ac.kr

**Keywords:** delirium, prevention, quetiapine, medical intensive care unit, extubation

## Abstract

Purpose: To evaluate the efficacy of short-term low-dose quetiapine for delirium prevention in critically ill patients. Methods: In this prospective, a single-center, randomized, double-blind, placebo-controlled trial, adult patients who were admitted from July 2015 to July 2017 to a medical intensive care unit (ICU) of a tertiary teaching hospital affiliated to Seoul National University were included. Quetiapine (12.5 mg or 25 mg oral at night; *N* = 16) or placebo (*N* = 21) was administered according to randomization until ICU discharge or the 10th ICU day. The primary endpoint was the incidence of delirium within the first 10 ICU days. Secondary endpoints included the rate of positive Confusion Assessment Method for the ICU (CAM-ICU) (the number of positive CAM-ICU counts/the number of total CAM-ICU counts), delirium duration, successful extubation, and overall mortality. Result: The incidence of delirium during the 10 days after ICU admission was 46.7% (7/15) in the quetiapine group and 55.0% (11/20) in the placebo group (*p* = 0.442). In the quetiapine group, the rate of positive CAM-ICU was significantly lower than in the placebo group (14.4% vs. 37.4%, *p* = 0.048), delirium duration during the study period was significantly shorter (0.28 day vs. 1.83 days, *p* = 0.018), and more patients in the quetiapine than in the placebo group were weaned from mechanical ventilation successfully (84.6% vs. 47.1%, *p* = 0.040). Conclusions: Our study suggests that prophylactic use of low-dose quetiapine could be helpful for preventing delirium in critically ill patients. A further large-scale prospective study is needed.

## 1. Introduction

Delirium is defined as a serious disturbance of cognition and consciousness by medical illnesses or neurologic causes by DSM-V criteria [1,2,3] and occurs in up to 30–80% of patients in the intensive care unit (ICU) [4,5,6,7]. The clinical presentation of delirium is often characterized by acute onset, an altered level of consciousness, a fluctuating course, and a global impairment of cognitive functioning. The development of delirium leads to a longer duration of an ICU or hospital stay and to increased mortality [4,8,9]. In terms of survival after hospital discharge, delirium is also known as a major risk factor for long-term sequelae, such as cognitive impairment and physical disability [10,11,12].

Clinical practice guidelines for pain, agitation, and delirium were published in 2018, and emphasize that the use of delirium assessment tools to monitor ICU patients can improve treatment effects due to a more rapid delirium detection and treatment [13]. According to 2018 guidelines [13], pharmacologic delirium prophylaxis was not recommended in adult ICU patients due to the lack of supporting clinical data regarding reduced delirium incidence or prevalence. However, since the guidelines have been published, studies on the pharmacologic prevention of delirium in ICU patients have been reported. In these reports, haloperidol [14,15,16,17,18] has been studied most frequently, and RCTs using rivastigmine [19], risperidone [20], and dexmedetomidine [21,22] have also been published.

Quetiapine is an atypical antipsychotic that is generally used to manage neuropsychiatric diseases such as depression or schizophrenia [23]. One prospective study showed the treatment efficacy of quetiapine for ICU delirium; quetiapine reduced the time to the first resolution of delirium (1.0-day vs. 4.5 days, *p* = 0.001) and the duration of delirium (1.5 days vs. 5 days, *p* = 0.006) in the study versus control group [24]. However, no study has yet been conducted to determine whether prophylactic use of quetiapine can prevent delirium in ICU patients. Therefore, we aimed to evaluate the efficacy of low-dose quetiapine for preventing delirium in critically ill patients.

## 2. Materials and Methods

### 2.1. Study Design

A prospective, randomized, double-blind, placebo-controlled study was conducted in a medical intensive care unit (MICU) of Seoul National University Bundang Hospital (SNUBH) from July 2015 to July 2017. This study was conducted in accordance with the amended Declaration of Helsinki. The Institutional Review Board (IRB) of SNUBH approved the study protocol, and all enrolled patients or their caregivers gave written informed consent before the randomization (IRB no.: B-1404-247-009). In response to our investigational drug application for this investigator-initiated trial, the Korea Food and Drug Administration approved the trial. Trial Registration: The trial was registered online before recruitment started (NCT02297763). Registered 1 July 2015, https://clinicaltrials.gov/ct2/show/NCT02297763?term=NCT02297763&rank=1.

### 2.2. Enrollment

During the study period, adult patients admitted to the MICU were screened for eligibility. They were eligible for the study when three or more of the following inclusion criteria were met and none of the exclusion criteria were applicable. The inclusion criteria were as follows: age over 64 years, acute physiology and chronic health evaluation II (APACHE-II) score over 14 points, suspicion of infection, intubation and mechanical ventilation, receiving continuous renal replacement therapy, ongoing metabolic acidosis, use of morphine or sedatives, unexpected ICU admission, and non-sustained coma (drug-related and others).

We excluded patients from enrolment for the following reasons: age less than 18 years old, current pregnancy, delirious at the time of ICU admission (initial CAM-ICU (the Confusion Assessment Method-ICU) positive), could not communicate within 3 months of ICU admission due to previously diagnosed irreversible neurologic disease (stroke, cerebral hemorrhage, traumatic brain injury, recent brain surgery, severe dementia, etc.), acute neurologic disease or injury at ICU admission, hepatic encephalopathy or liver cirrhosis with a Child-Pugh score B or C, ongoing outpatient or inpatient anti-psychotic drug use, high risk for ventricular arrhythmias (ongoing treatment with drugs known to prolong the QT interval (e.g., erythromycin, class Ia, Ic, III anti-arrhythmic drugs), high risk for drug interaction with quetiapine (phenytoin, carbamazepine, barbiturates, proteinase inhibitors, nefazodone), use of central nervous system inhibitory drugs barbiturates), epinephrine, severe bradycardia, hematologic malignancy, suspected death within 72 h of ICU admission, and refusal of informed consent.

### 2.3. Randomization

After informed consent, patients were randomized to either placebo or quetiapine group within 72 h of ICU admission. Randomization was performed by the hospital pharmacist using the randomization table made by a biostatistician in the Medical Research Collaborating Center of the hospital. Without the pharmacist, neither study personnel nor patients were aware of the treatment group assignment.

### 2.4. Study Protocols

After randomization, patients received one of the two study drugs: 12.5 mg or 25 mg quetiapine (as a solution of 10 mL), or placebo (same amount of starch powder as a solution of 10 mL) (Figure 1). Quetiapine administration (by mouth; par os) was started at 12.5 mg and could be increased up to 25 mg at 24-h intervals if there were no adverse reactions. We used 25 mg quetiapine tablets as the study drug. To apply 12.5 mg, a quetiapine 25 mg tablet was crushed, and the powder was dissolved in sterilized distilled water to make a 20 mL solution. Half of this volume was then used as a 12.5 mg quetiapine/10 mL solution. To administer 25 mg, the powder of a 25 mg quetiapine tablet was dissolved in sterilized distilled water to prepare a 10 mL solution (25 mg quetiapine/10 mL solution). For placebo, the same amount of starch powder was dissolved in sterilized distilled water to produce 10 mL of the placebo solution. Both quetiapine and placebo were provided in identical vials and wrapped in a white label to conceal the contents. Thus, the entire study team, enrolled patients, and clinicians were blinded during the study procedure. The study drug was delivered by the study nurse from the hospital pharmacy to the ICU, and it was administered by mouth or via a nasogastric tube at night, once every 24 h. During the study period, enrolled patients were examined three times per day for signs of delirium using the CAM-ICU. According to the ICU guidelines of SNUBH, other sedatives (dexmedetomidine, midazolam), pain killers (fentanyl, remifentanil), or delirium treating drugs (haloperidol, lorazepam, other atypical anti-psychotics that were not quetiapine) were administered when indicated. During the course of the study, the study nurse examined the patients for possible adverse events (QT prolongation, hypotension, bradycardia, elevated liver enzymes, extrapyramidal symptoms, etc.) related to the drug administration.

### 2.5. Outcomes and Data C ollection

The primary predefined outcome was delirium incidence diagnosed with the CAM-ICU within the study period. If the CAM-ICU was positive at least once during the study period, it was recorded that a delirium occurred. Secondary outcomes were the rate of positive CAM-ICU (the number of positive CAM-ICU counts/the number of total CAM-ICU counts), the duration of delirium within the 10th ICU day or ICU discharge day (in cases when ICU discharge occurred before the 10th ICU day), the length of stay in the ICU, the length of stay in the hospital, the duration of intubation, a successful extubation, the ICU mortality, the overall mortality, and the use of rescue medication.

### 2.6. Statistical Analyses

To determine the effectiveness of quetiapine for ICU delirium prevention, we used the chi-square test, a two-tailed test with delirium incidence and baseline characteristics with an 80% power and a 5% significance level between two groups. Based on a preliminary test in 2012 at SNUBH, the delirium incidence at the MICU was about 75%. Additionally, other trials of delirium prevention have shown a 7–20% delirium decrease with haloperidol or risperidone prophylaxis [13,14,19]. We aimed for a 20% difference between the study and placebo groups and took a 10% drop-out rate into consideration; we, therefore, calculated that at least 97 patients would be required in each group.

Data were analyzed using SPSS software for Windows, version 22.0 (IBM Corp., Armonk, NY, USA). The participants’ clinical variables were analyzed using descriptive statistics. The results are expressed as mean and standard deviation. Variables were compared between the study and control groups. Categorical variables were compared by chi-square analysis and continuous variables were compared using Student’s *t*-test or the Mann–Whitney test. ICU delirium incidence, successful extubation, ICU mortality, and in-hospital mortality were shown as a forest plot. For all analyses, *p* values < 0.05 were considered significant.

## 3. Results

### 3.1. Baseline Characteristics

During the intervention period (from July 2015 to July 2017), 205 patients were screened and 37 fulfilled the inclusion criteria (Online supplement, Figure 2). After the inclusion of 37 patients, the recruitment period was ended early because the study drug was close to the expiration date. Of the 37 patients, 16 patients were assigned to the quetiapine group and 21 received placebos. After one patient dropped out from both groups, 15 patients were included in the study group and 20 in the control group. Between the placebo group and the study group, there were no significant differences in baseline characteristics (Table 1).

### 3.2. Clinical Outcomes Related to Delirium in ICU

Our primary outcome, the overall incidence of delirium during 10 days of ICU admission, was 46.7% (*N* = 7) in the quetiapine group and 55.0% (*N* = 11) in the placebo group (*p* = 0.442). However, the rate of positive CAM-ICU was significantly lower in the quetiapine group compared to the placebo group (14.4% vs. 37.5%, *p* = 0.048). Moreover, delirium duration during the study period was significantly shorter in the study group than in the control group (0.28 day vs. 1.83 days, *p* = 0.018), and more patients in the quetiapine group were successfully weaned from mechanical ventilation (*N* = 11, 84.6% vs. *N* = 8, 47.1%, *p* = 0.040). Regarding ICU or in-hospital mortality, there was no difference between groups (Table 2). However, the quetiapine group showed more favorable outcomes regarding ICU delirium incidence (odds ratio (OR), 0.72; 95% confidence interval (CI), 0.19–2.74), successful extubation (OR, 0.16; 95% CI, 0.03–0.96), ICU mortality (OR, 0.36; 95% CI, 0.06–2.11), and in-hospital mortality (OR, 0.25; 95% CI, 0.05–1.17) (Figure 3). The amount of analgesics and sedatives used during the study period after randomization were not different between the two groups (Appendix A). In addition, haloperidol, lorazepam, and amisulpride were used in 12 patients, but there was no significant difference between the two groups. During administration of the study drug, adverse events such as QT prolongation, hypotension, bradycardia, elevated liver enzymes, or extrapyramidal symptoms were not observed.

## 4. Discussion

To the best of our knowledge, this is the first prospective, randomized, double-blind, placebo-controlled study to evaluate the effectiveness of prophylactic quetiapine for delirium prevention in critically ill patients in medical ICUs. Although the primary outcome of a statistically significant 20% reduction in ICU delirium incidence was not observed, the quetiapine group showed favorable results relative to the control group; these results were a lower CAM-ICU positive rate, shorter duration of delirium in the first 10 ICU days, and a higher success rate for mechanical ventilation weaning. Therefore, our study suggests that low-dose quetiapine could play a potential role in preventing ICU delirium.

Benzodiazepine or narcotic analgesics administered for sedation and pain control are well known risk factors [25,26] for delirium, thus, it is recommended that these drugs be used as little as possible for this purpose [27,28]. Additionally, it is important to minimize the risk of delirium such as restraining (anchoring) the patient on the bed, as well as the environmental problems of the ICU such as sleep disturbance through various interventions during nighttime. Due to the varied causes of delirium, effective prevention relies on the integrated control of different risk factors, requiring collaborative effort from ICU staff through bundled care. However, preventing delirium though pharmacologic agents is an attractive possibility due to the relatively simple administration of drugs compared to the implementation of non-pharmacologic interventions, such as daily awakening, early rehabilitation exercise, improving the ICU environment, and following sleep-enhancing protocols, which are more time consuming for ICU staffs.

The development of effective drugs for preventing delirium should be based on the pathophysiologic mechanism by which delirium develops. This mechanism, however, especially when associated with a critical illness, remains largely unknown, and there is currently no medication approved by the US Food and Drug Administration for the prevention of delirium. Delirium research using neuroimaging has provided evidence that delirium may be caused by widespread brain dysfunction rather than localized disruption [29], and studies using magnetic resonance imaging have shown that delirium in ICU patients is associated with anatomical alternations such as brain atrophy and brain white matter rupture [30,31]. Furthermore, the incidence of delirium increases with the use of γ-aminobutyric acid antagonists and anticholinergic drugs. These drugs have been reported to cause delirium by various mechanisms such as decreased cholinergic neurotransmitter levels, excessive dopamine activity, and direct neurotoxicity of proinflammatory cytokines [32].

The study drug, quetiapine, which is an atypical antipsychotic drug, is primarily used in major depressive or general anxiety disorders. Since quetiapine has a higher affinity for 5-HT2 receptors rather than D2 receptors, it causes less extrapyramidal symptoms, and it also has a sedative effect due to a suppressive action via H1 receptors [33]. Therefore, it has already been selected as a treatment option to manage ICU delirium, and a post-hoc study suggested that quetiapine is effective and resolves delirium symptoms faster than placebo (4 days vs. 14 days, *p* = 0.004) [34]. Another prospective, randomized, placebo-controlled pilot study showed that quetiapine decreased the time to the first delirium resolution (1.0-day vs. 4.5 days, *p* = 0.001), reduced the delirium duration (36 h vs. 120 h, *p* = 0.006), and caused less agitation (Sedation-Agitation Scale score ≥ 5, 6 h vs. 36 h, *p* = 0.02) [24].

The present study used quetiapine prophylactically in medical ICU patients and demonstrated several meaningful results. Although quetiapine did not decrease ICU delirium incidence, it had an impact on the CAM-ICU positive rate (quetiapine vs. placebo, 14% vs. 37%) and reduced the duration of delirium. This means that the patients in the quetiapine group had less fluctuation in delirious symptoms. Additionally, the quetiapine group had a higher success rate of being weaned from mechanical ventilation. Although hospital mortality was not statistically different between groups (quetiapine vs. placebo, 20% vs. 50%, *p* = 0.070), the study group showed more favorable results according to the forest plot (Figure 3).

There are several evidence-based hypotheses that could explain the findings of the present study. First, a lower CAM-ICU positive rate and shorter delirium duration of the quetiapine group would make it possible to more successfully wean the study group patients from mechanical ventilation compared to the control group, and the significant difference found in this study could have influenced the favorable results in mortality. Second, it is possible that quetiapine has anti-inflammatory effects. Thus, it is possible that the favorable outcomes reported in this study were caused by anti-inflammatory effects of the drug rather than by the prevention of delirium. The authors of two experimental studies suggest an anti-inflammatory potency of quetiapine [23,35]. According to Jaehne et al., quetiapine increases anti-inflammatory and decreases pro-inflammatory cytokine levels in mice [35]. Additionally, in a murine collagen-induced arthritis (CIA) model, the concentrations of anti-type II collagen-specific antibodies, interleukin-6, interleukin-17, and prostaglandin E_2_ were significantly lower in the quetiapine-treated group compared to the control group (*p* < 0.05) [23]. Although the anti-inflammatory effects of quetiapine have been verified by clinical trials, the possibility that this effect influenced our favorable outcomes cannot be completely ruled out.

To date, nine randomized controlled trials (RCTs) for pharmacologic delirium prevention in the ICU setting have been published; five of these were studied with haloperidol, two with dexmedetomidine, and one with risperidone and rivastigmine (Appendix A). Among these nine RCTs, six were designed for postoperative settings (Appendix A). There have been three trials [17,18,22] that included critically ill patients in medical ICUs as well as in surgical and trauma ICUs. The MIND trial [17] compared three groups: haloperidol, ziprasidone, and placebo; the results showed no difference in delirium-related outcomes in medical, surgical, or trauma ICUs [17]. In the REDUCE trial [18] involving 1789 critically ill patients in medical, surgical, or trauma ICUs, there was no significant difference in the number of days survived at 28 days following inclusion between patients who received prophylactic haloperidol (1 mg or 2 mg) and patients who received placebo. The other trial was published recently by Yoanna et al. [22], and they showed that nocturnal administration of low-dose dexmedetomidine in critically ill adults reduced the incidence of delirium during the ICU stay. They hypothesized that nocturnal dexmedetomidine would prevent delirium by improving sleep quality, but patient-reported sleep quality appeared unchanged in general medical/surgical critically ill adults.

The present study differs from these trials in that only patients in a medical ICU were enrolled. Most RCTs on pharmacologic prevention of delirium in the ICU included only postoperative patients or mixed ICU patients. These populations are usually in less severe states than patients in a medical ICU and rarely require long-term ICU management. Therefore, findings from the previous studies were not generalizable to critically ill patients in medical ICUs. Although our study population was relatively small, our findings, which suggest promising efficacy of low-dose quetiapine in preventing ICU delirium, are noteworthy. Future prospective multicenter RCTs are needed to confirm our results. Additional strengths of our study were the prospective design and the nearly complete blinding that was accomplished by the white-labeled drug and placebo doses. Finally, we tested a very low dose (12.5 mg or 25 mg) of quetiapine that was a much smaller amount than the usual dosage, and as a result, concerns of side effects could be minimized. None of the patients in the quetiapine group experienced any adverse events related to the study drug.

There are several limitations to our study. First, the number of patients in our study was small, which limits an extended interpretation of the results. The strict study inclusion/exclusion criteria led to only 17% of screened patients being enrolled, potentially compromising external validity. As mentioned in the Methods, we terminated this study earlier than planned due to difficulties in patient enrolment and the expiration date of the study drug. Due to the severity of the patients’ illnesses in medical ICUs, it was difficult to obtain consent from the caregivers, especially within 72 h of ICU admission, which was before concrete rapport could be formed between ICU staff, patients, and caregivers. Second, this study did not elucidate the precise mechanism of the favorable outcomes shown in the quetiapine group, especially regarding the drug’s effects on mortality. Although we presumed that the favorable outcomes in mortality were caused by the influence of the drug on delirium (fewer fluctuations and shorter duration), the effect might have been caused by an anti-inflammatory effect of quetiapine. However, we could not verify this mechanism because we did not collect patient blood samples. Third, as our study lasted for two years, ICU practice might have changed over a two-year period, which may have affected our findings. Finally, we did not follow up with patients to observe whether delirium developed after ICU discharge, or to determine other long-term outcomes after hospital discharge.

## 5. Conclusions

Our study suggests that prophylactic use of low-dose quetiapine could be helpful for the prevention of delirium in critically ill patients. Although the incidence of ICU delirium did not decrease, patients treated with quetiapine in this study experienced less fluctuation of symptoms, shorter durations of delirium, and a higher success rate of being weaned from mechanical ventilation. To support these beneficial effects of quetiapine through additional evidence, a large-scale prospective study is needed.

## Figures and Tables

**Figure 1 jcm-09-00069-f001:**
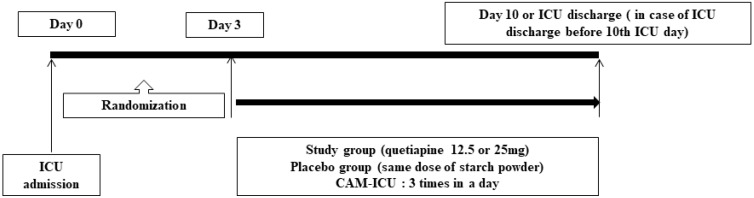
Study protocol.

**Figure 2 jcm-09-00069-f002:**
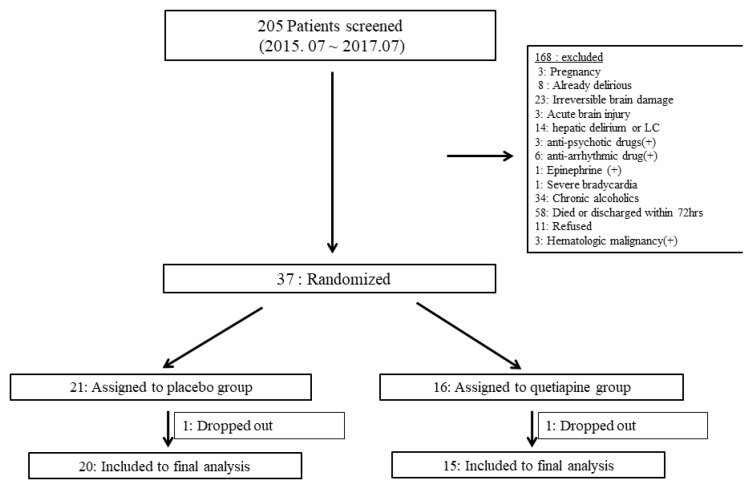
Enrolment and randomization of the study participants.

**Figure 3 jcm-09-00069-f003:**
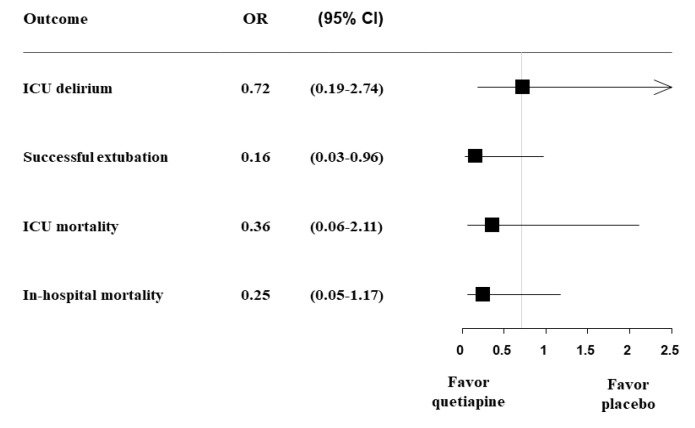
Forest plot of the delirium-related outcomes.

**Table 1 jcm-09-00069-t001:** Baseline characteristics of enrolled patients.

Variables	Placebo Group (*N* = 20)	Study Group (*N* = 15)	*p* Value
Age	69.10 ± 12.42	71.33 ± 10.35	0.576
Sex, male	11 (55.0)	11 (73.3)	0.226
APACHE II score	23.65 ± 7.85	21.53 ± 10.15	0.491
SOFA score	7.10 ± 4.08	5.60 ± 4.12	0.291
Charlson comorbidity index	4.70 ± 2.41	6.07 ± 1.94	0.081
Intubated at study entry	17 (85.0)	13 (86.7)	0.64
Baseline QTc interval, ms	449.25 ± 36.68	457.07 ± 30.85	0.510
Mean RASS at study entry	−1.83 ± 2.12	−1.43 ± 1.53	0.549
Cause of ICU admission			0.659
Pneumonia (+ARDS)	5 (25.0)	6 (40.0)	
Respiratory failure	13 (65.0)	8 (53.3)	
Cardiogenic origin	1 (5.0)	1 (6.7)	
Others	1 (5.0)	0 (0.0)	
Sepsis/Septic shock	6 (30.0)	3 (20.0)	0.700
Sedative, analgesics within 24 h before randomization			
Dexmedetomidine (mcg)	246.40 ± 326.87	213.40 ± 419.83	0.795
Midazolam (mg)	23.24 ± 44.32	14.73 ± 28.95	0.523
Remifentanil (mg)	2.26 ± 1.98	2.17 ± 2.38	0.902

Note: Data are expressed as number (percentage) or mean ± standard deviation. Abbreviations: APACHE II score, Acute Physiology and Chronic Health Evaluation II score; SOFA; Sequential Organ Failure Assessment; ICU, intensive care unit; RASS, Richmond Agitation-Sedation Scale; ARDS, acute respiratory distress syndrome.

**Table 2 jcm-09-00069-t002:** Clinical outcomes related to ICU delirium.

Variables	Placebo Group (*N* = 20)	Study Group (*N* = 15)	*p* Value
Delirium incidence	11 (55.0)	7 (46.7)	0.442
Delirium subtype			0.431
Hypoactive (−5–−1)	4 (20.0)	1 (6.7)	
Hyperactive (0–+4)	5 (25.0)	6 (40.0)	
Mixed (−5–+4)	11 (55.0)	8 (53.3)	
Positive rate of CAM-ICU	0.37 ± 0.38	0.14 ± 0.28	0.048
Duration of delirium (days)	1.83 ± 1.34	0.28 ± 0.52	0.018
Drug administration days	5.40 ± 1.70	3.93 ± 2.02	0.260
Average dose of study drug (mg/day)	21.42 ± 2.24	20.40 ± 2.72	0.233
Time spent agitated, RASS > +2–+4			
Hours	7.80 ± 13.64	7.20 ± 8.84	0.883
Percent	0.05 ± 0.08	0.06 ± 0.08	0.542
Time spent deeply sedated, RASS < −2–−4			
Hours	71.40 ± 80.31	32.00 ± 45.73	0.076
Percent	0.35 ± 0.32	0.24 ± 0.29	0.278
Patient initiated device removal			
Endotracheal tube	0	0	-
C lines, A line, or IV line	1 (5.0)	2 (13.3)	0.383
Levin tube	4 (20.0)	4 (26.7)	0.642
MV apply	17 (85.0)	13 (86.7)	0.640
Duration of MV (days)	15.76 ± 24.22	6.43 ± 8.24	0.180
ICU LOS (days)	17.00 ± 22.56	7.47 ± 7.31	0.126
Hospital LOS (days)	35.25 ± 29.60	25.33 ± 21.84	0.283
Successful extubation	8 (47.1)	11 (84.6)	0.034
ICU mortality	6 (30.0)	2 (13.3)	0.245
In-hospital mortality	10 (50.0)	3 (20.0)	0.070
Discharge to			0.185
Home	7 (35.0)	9 (60.0)	
Chronic facility care	3 (15.0)	3 (20.0)	
Death	10 (50.0)	3 (20.0)	

Note: Data are expressed as number (percentage) or mean ± standard deviation. Abbreviations: CAM-ICU, Confusion Assessment Method-Intensive Care Unit; ICU, intensive care unit; RASS, Richmond Agitation-Sedation Scale; C-line, Central line; A-line, arterial line; IV line, intravenous line; MV, mechanical ventilation; LOS, length of stay.

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
