# Peer review of "Efficacy of Low-Dose Prophylactic Quetiapine on Delirium Prevention in Critically Ill Patients: A Prospective, Randomized, Double-Blind, Placebo-Controlled Study"

_jcm, 2019, doi:10.3390/jcm9010069_

Round 1

Reviewer 1 Report

With much interest I have read the MS Efficacy of low-dose prophylactic quetiapine on delirium prevention in critically ill patients: A prospective, randomized, double-blind, placebo-controlled study

The study investigates the possible rols of quetiapine in preventig delirium in critically ill patients.

The aim of the study is clear.

Vedi come hanno dato la quetiapina?

Unfortunately the study has enrolled a very limited number of patients and I wonder whether this study has enough statistical power. Is power calculation necessary?

The authors conclude that the findings of the study suggest that prophylactic us of low dose quetiapine may be helpful in preventing delirium. I believe this informaion is misleading and these statements must be calibrate thruoghout the MS. Actually low-dose quetiapine is as efficient as placebo in preventing delirium.

Information should be given on the way quetiapine was administrated. Furthermore the authors should show information on existing cognitive disorders in patinets of both groups. Cognitive disorders might have influenced the outcomes!

Secondary endpoints are in favour of the treated group, actually these results should be stressed in the conclusions.

The discussion is very long. I suggest the authors to stick at their findings within the framework of the limitations of the study. Therefore, I think that any conclusion is relatively specultative. This must be pointed out very well.

Author Response

Response to the Reviewer #1’s comments:

With much interest I have read the MS Efficacy of low-dose prophylactic quetiapine on delirium prevention in critically ill patients: A prospective, randomized, double-blind, placebo-controlled study.

The study investigates the possible roles of quetiapine in preventing delirium in critically ill patients.

The aim of the study is clear.

Response: Thank you for your positive comments.

Comment 1: Unfortunately, the study has enrolled a very limited number of patients and I wonder whether this study has enough statistical power. Is power calculation necessary?

Response: We agree with your concern. Although the number of enrolled patients in this study is not statistically strong enough, the secondary outcome such as positive rate of CAM-ICU, duration of delirium, successful extubation, ICU mortality, in-hospital mortality showed the favorable result in quetiapine group. Therefore, we think that low-dose quetiapine might play a role in preventing delirium. In the future, a large-scale prospective study will be needed to prove this.

Comment 2: The authors conclude that the findings of the study suggest that prophylactic use of low dose quetiapine may be helpful in preventing delirium. I believe this information is misleading and these statements must be calibrate throughout the MS. Actually low-dose quetiapine is as effective as placebo in preventing delirium.

Response: Thank you for your valuable comment. As you commented, we have corrected the conclusion of our manuscript.

Comment3: Information should be given on the way quetiapine was administrated.

Response: The way quetiapine was administrated showed in the study protocols at Methods section (page 5, line 116-135).

Comment 4: Furthermore, the authors should show information on existing cognitive disorders in patients of both groups. Cognitive disorders might have influenced the outcomes.

Response: Thank you for comments. In the beginning of the study, we excluded delirious patients at the time of ICU admission (initial CAM-ICU [the Confusion Assessment Method-ICU] positive), patients who could not communicate within 3 months of ICU admission due to previously diagnosed irreversible neurologic disease(stroke, cerebral hemorrhage, traumatic brain injury, recent brain surgery, severe dementia, etc.), acute neurologic disease or injury at ICU admission, hepatic encephalopathy with a Child-Pugh score B or C, ongoing outpatient or inpatient anti-psychotic drug use. We described this in Enrollment of Methods section (page 4, line 98-102).

Comment4: Secondary endpoints are in favor of the treated group, actually these results should be stressed in the conclusions.

Response: We agree with your comment and we have corrected our conclusions. 

Comment5: The discussion is very long. I suggest the authors to stick at their findings within the framework of the limitations of the study. Therefore, I think that any conclusion is relatively specultative. This must be pointed out very well.

Response: Thank you for your comments, again. As you recommended, we have deleted a few paragraphs from the discussion and have corrected the conclusions of our manuscript.  

Reviewer 2 Report

Although the author have to be commended for their work, this is a heavily underpowered study to assess effectiveness of prophylaxis with quetiapine for ICU delirium at a medical ICU, and should be handled as such by the authors (in general their results have now been overstated in terms of clinical value, given the too small size of the population recruited).

Specific comments;

-Introduction: 1)stick with DSM-V to describe delirium. 2)Do not refer to the PAD 2013 guidelines but to the 2018 PADIS guidelines (this is the updated version). 

-Methods: these are insufficient, especially for an RCT. Adhere strictly to the CONSORT criteria for RCTs in the description and add a checklist detailing the RCT criteria that should have been described. Describe the methods in manuscript instead of the online supplement!. 

-Results: describe better why recruitment fell behind so heavily.

-The primary outcome: delirium incidence, is mentioned as non significant in first sentence of 3.2 (p=0.442), but a few sentences later it is stated: " However, the quetiapine group.showed more favourable outcomes regarding ICU delirium incidence..". These findings seem contradictory: please explain this.

In general the conclusions are heavily overstated: you did NOT find a  proven effect, since the trial was underpowered, therefore, strong conclusions are not appropriate. Please tone down your description of the results throughput the manuscript. Regard this trial as an underpowered trial, showing some significant results that may more likely be due to low numbers (and coincidence) or case mix differences between the groups, than true effects. Therefore regard any significant results as being hypothesis generating at most, rather than showing any proof for efficacy. Also, some effects seem too strong for such low numbers (extubation with significant OR of 0.16, indicating a very strong effect is highly unlikely due to the intervention, but also due to the very low numbers of randomised patients: please check/re-check these results since they have a high likelihood of being false in my view given extreme effect-measure.

Author Response

Response to the Reviewer #2’s comments:

Although the author have to be commended for their work, this is a heavily underpowered study to assess effectiveness of prophylaxis with quetiapine for ICU delirium at a medical ICU, and should be handled as such by the authors (in general their results have now been overstated in terms of clinical value, given the too small size of the population recruited).

Response: We agree with your concern. Although the number of N in this study is not statistically strong enough, the secondary outcome such as positive rate of CAM-ICU, duration of delirium, successful extubation, ICU mortality, in-hospital mortality showed the favorable result in quetiapine group. Therefore, we think that low-dose quetiapine might play a role in preventing delirium. In the future, a large-scale prospective study will be needed to prove this.

Specific comments:

Introduction)

Stick with DSM-V to describe delirium.

Response: Thank you for your comment. We have described the delirium following DSM-V criteria.

“Delirium is defined as a serious disturbance of cognition and consciousness by medical illnesses or neurologic causes by DSM-V criteria” (Page 3, line 55-56).

Do not refer to the PAD 2013 guidelines but to the 2018 PADIS guidelines (this is the updated version).

Response: We have updated as the 2018 PADIS guidelines instead of the PAD 2013 guidelines (page 3, line 62-64).

Methods) These are insufficient, especially for an RCT. Adhere strictly to the CONSORT criteria for RCTs in the description and add a checklist detailing the RCT criteria that should have been described. Describe the methods in manuscript instead of the online supplement.

Response: We have checked and added the CONSORT criteria for our RCT trial at the supplementary file. Additionally, we have described the detailed methods in the Methods section.

Results) Describe better why recruitment fell behind so heavily.

Response: We have described the reasons why recruitment fell behind so heavily in Discussion section. First, the study passed the expiration date. And we had to terminate this study earlier than planned due to difficulties in patient enrollment. Due to the severity of the patients’ illnesses in medical ICUs, it was difficult to obtain consent from the caregivers, especially within 72 hours of ICU admission, which was before concrete rapport could be formed between ICU staff, patients, and caregivers (page10, line 277-280).

The primary outcome: delirium incidence, is mentioned as non-significant in first sentence of 3.2 (p=0.442), but a few sentences later it is stated: “However, the quetiapine group showed more favorable outcomes regarding ICU delirium incidence.”. These findings seem contradictory; please explain this.

Response: In the beginning of the study, we had considered ICU delirium incidence as the primary outcome, however, the ICU delirium incidence did not show the statistical important. Although quetiapine did not decrease ICU delirium incidence, it showed the significant difference on the CAM-ICU positive rate (quetiapine vs. placebo, 14% vs 37%) and reduced the duration of delirium. This means that the patients in the quetiapine group had less fluctuation in delirious symptoms. Additionally, the quetiapine group had a higher success rate of being weaned from mechanical ventilation (figure 3). From these findings, we described like that: “However, the quetiapine group showed more favorable outcomes regarding ICU delirium incidence.”.

In general, the conclusions are heavily overstated: you did NOT find a proven effect, since the trial was underpowered, therefore, strong conclusions are not appropriate. Please tone down your description of the results throughout the manuscripts.

Response: Thank you for your valuable comment. We have corrected the conclusion.

Regard this trial as an underpowered trial, showing some significant results that ma more likely be due to low numbers (and coincidence) or case mix differences between the groups, that true effects. Therefore, regard any significant results as being hypothesis generating at most, rather than showing any proof for efficacy.

Response: Thank you for your precious comments. As you pointed out, the secondary outcomes such as successful extubation, in-hospital mortality or ICU mortality showed very dramatic differences between study group and placebo group. Because this study has finished and analyzed with a small number of patients, we have revised our manuscript by presenting the possibility of the quetiapine’s role in prevention, rather than by expressing the effect of quetiapine.

Also, some effects seem too strong for such low numbers (extubation with significant OR of 0.16, indicating a very strong effect is highly unlikely due to the intervention, but also due to the very low numbers of randomized patients: please check/re-check these results since they have a high likelihood of being false in my view given extreme effect-measure.

Response: Thank you for your precious comments. The researcher who analyzed the results did not know the blinding group in the process of collecting results.